# Predicting Affinity Through Homology (PATH): Interpretable binding affinity prediction with persistent homology

**Yuxi Long[1], Bruce R. Donald** [1,2]*

**1** Department of Computer Science, Department of Mathematics, Duke University, Durham, North Carolina, United States of America, **2** Department of Biochemistry, Department of Chemistry, Duke University and Duke University School of Medicine, Durham, North Carolina, United States of America

* brd+pcb25@cs.duke.edu

**Data availability statement:** Source code for inferencing with PATH+ and PATH- is located at https://github.com/donaldlab/OSPREY3/tree/main/src/main/python/path. Training source

## Abstract

Accurate binding affinity prediction (BAP) is crucial to structure-based drug design. We present *PATH*⁺, a novel, generalizable machine learning algorithm for BAP that exploits recent advances in computational topology. Compared to current binding affinity prediction algorithms, PATH⁺ shows similar or better accuracy and is more generalizable across orthogonal datasets. PATH⁺ is not only one of the most accurate algorithms for BAP, it is also the first algorithm that is inherently interpretable. Interpretability is a key factor of trust for an algorithm and alongside generalizability, which allows PATH⁺ to be trusted in critical applications, such as inhibitor design. We visualized the features captured by PATH⁺ for two clinically relevant protein-ligand complexes and find that PATH⁺ captures binding-relevant structural mutations that are corroborated by biochemical data. Our work also sheds light on the features captured by current computational topology BAP algorithms that contributed to their high performance, which have been poorly understood. PATH⁺ also offers an improvement of $\mathcal{O}(m+n)^3$ in computational complexity and is empirically over 10 times faster than the dominant (uninterpretable) computational topology algorithm for BAP. Based on insights from PATH⁺, we built PATH⁻, a scoring function for differentiating between binders and non-binders that has outstanding accuracy against 11 current algorithms for BAP. In summary, we report progress in a novel combination of interpretability, speed, and accuracy that should further empower topological screening of large virtual inhibitor libraries to protein targets, and allow binding affinity predictions to be understood and trusted. The source code for PATH⁺ and PATH⁻ is released open-source as part of the OSPREY protein design software package.

## Author summary

Predicting how strongly a small molecule (ligand) binds to a protein is a fundamental challenge in drug discovery. Recently, deep learning methods have shown promise in

 

code for PATH+ and PATH- is located at https://github.com/longyuxi/PATH-training. Source code for the open-source implementation of TNet-BP is available at https://github.com/longyuxi/TopologyNet-2017. The PDBBind dataset can be found at http://pdbbind.org.cn/. The BioLiP dataset (containing BindingDB and Binding MOAD) can be found at https://zhanggroup.org/BioLiP/weekly.html and downloaded with https://zhanggroup.org/BioLiP/download/download_all_sets.txt. The DUD-E dataset can be found at https://dude.docking.org/, and we additionally provide a mirror of the DUD-E dataset at https://doi.org/10.6084/m9.figshare.29132615.

**Funding:** This research was supported by National Institute of Health (NIH, https://www.nih.gov/) grant R35 GM-144042 to B.R.D. NIH did not play any role in the study design, data collection and analysis, decision to publish, or preparation of the manuscript.

**Competing interests:** I have read the journal's policy and the authors of this manuscript have the following competing interests: B.R.D. is a founder of Ten63 Therapeutics, Inc. B.R.D. was previously a guest editor for PLoS Comp. Biol.

this task. However, we find that many of these models suffer from **overfitting**, meaning they perform well on their training data but fail to generalize to new datasets. This is concerning because practical drug discovery requires models that work well beyond their training set. Additionally, most previous algorithms—including both deep learning and traditional methods—**overestimate** binding affinity and predict that most protein-ligand pairs interact favorably, when in reality the vast majority of molecules do not bind to their targets at all. To address these challenges, we introduce **PATH⁺**, a new algorithm that encodes structural binding features using **persistent homology**, a mathematical tool from algebraic topology. Our **persistence fingerprint** efficiently captures geometric properties such as molecular cavities and interaction patterns at multiple scales. PATH⁺ significantly **outperforms** previous affinity prediction methods on unseen data while being **interpretable**—meaning predictions can be traced back to specific atomic interactions. Additionally, we develop **PATH⁻**, a scoring function that improves discrimination between true binders and non-binders. Finally, we provide a **provably accurate** algorithm that improves the efficiency of persistent homology computations by a cubic factor, making PATH ten times faster than previous topology-based methods. Our work advances both **computational topology** and **in silico drug discovery**, improving accuracy, efficiency, and interpretability in binding affinity prediction.

## Introduction

Structure-based drug design (SBDD) is an invaluable tool for effective lead discovery [1]. An important step in SBDD is virtual screening, where large libraries of compounds are computationally screened against a protein target of known structure to predict inhibitors that bind to the target [2]. SBDD is enabled by docking, where a pose generation algorithm generates atomistic spatial conformations in which a given protein and ligand potentially bind, and a scoring algorithm selects the most promising conformations for further analysis [3,4]. A reliable predictor to discriminate binding versus non-binding docking poses and a ranking of good poses, based on affinity, potency, or other biophysical properties, should be important for accurate SBDD [4]. In this work, we present a powerful, novel pair of algorithms that not only classify binders versus non-binders, but also predict the binding affinity of a given protein-ligand conformation to set the order for experimental testing, computational redesign, or structure-based medicinal chemistry or diversification.

Binding affinity characterizes the strength of the interaction between a protein and a ligand. Binding affinity is also a key factor in determining the efficacy of a drug, as tight-binding ligands are more likely to be potent drugs *in vivo* [5–7]. Since experimental determination of protein-ligand binding affinity is time-consuming and costly in many cases [8], accurate methods for protein-ligand binding affinity prediction *in silico* are crucial in the structure-based drug design process [1,9]. One approach to *in silico* binding affinity prediction is through molecular dynamics simulations [10,11]. Unfortunately, while molecular dynamics is rigorous and accurate in predictions, it is computationally intensive [12] and is not suitable for virtual screening, where a large number of compounds must be screened. Therefore, many scoring functions [13,14] have been developed, including physics-based, regression-based, and knowledge-based scoring functions [14]. Recent years also saw the application of many deep learning (DL) techniques for binding affinity prediction, including convolutional neural networks [15–17], attention mechanism [5,18], and graph neural networks [19,20]. These deep learning methods have produced more accurate predictions than handcrafted scoring functions. Furthermore, binding compounds are rare. Less than 1% of the compounds

in a typical small molecule library will bind to a given protein [21], which makes classifying non-binding compounds necessary.

Beside accuracy, *interpretability* is an essential quality and a key factor of trust for an algorithm. A non-interpretable model, also called a *black box* model, can "predict the right answer for the wrong reason" [22]. As a result, it is questionable whether a black box model could generalize beyond the training dataset [22]. A prime example of this caveat is AlphaFold2 [23], a black box model that achieved near-experimental accuracy on the CASP14 protein folding challenge [24] but is unable to predict the impact of structure-disrupting mutations, which are frequently associated with protein aggregation, misfolding, and dysfunction [25, 26]. The inability to understand the underlying workings of a black box model means that such solecisms are hard to discover in advance and their causes can only be speculated. On the other hand, an interpretable model is more robust, since the model can be calibrated to adapt to scenarios or considerations outside of its training dataset [27], such as mutant proteins, which were not abundantly present in the PDB dataset. As a result, when a discrepancy between an interpretable algorithm and empirical measurements is discovered, its cause can be precisely identified and fixed. Furthermore, interpretations from a model can produce insights, contribute to the understanding of the underlying system [28], and facilitate development for further efficient algorithms. While physics-based scoring functions for binding affinity usually have some interpretability [14], the adoption of deep learning techniques poses an inherent challenge in interpretability in DL based algorithms [27].

A promising direction is topological data analysis, where the most prominent method is *persistent homology*. Persistent homology quantifies the shapes of protein-ligand complexes by computing the *persistence* of topological invariants like holes and voids in the biomolecular structures at different spatial resolutions [29]. We show that the features encoded by persistent homology can be both highly descriptive and interpretable (Section Persistent homology). Algorithms using persistent homology have been applied to neuronal morphologies [30] and protein cavity detection [31,32]. Persistent homology has been used in several algorithms for binding affinity prediction, where persistent homology features are combined with chemical features, and the prediction is made using a neural network [33–38]. Persistent homology shows promise as an approach for accurate binding affinity prediction, and an algorithm using persistent homology and deep learning won many challenges in the D3R Grand Challenge 3 [39,40].

For representing molecular features, persistent homology provides two advantages that correlate to known properties of biomolecules:

1. **Stability with respect to noise**. It has been proven that small changes in the input to persistent homology (measured by *bottleneck distance*) cause only small changes in the persistence diagrams (measured by *Gromov-Hausdorff distance*) [29,41], which are representations of persistent homology. Thus, in embedding biochemical structure, small differences in the protein structure that arise often due to the structural heterogeneity of proteins [42] will have little effect on the resulting representation.

2. **Invariance under translation and rotation**. The persistence diagram representation is invariant under translation and rotation of the biomolecule. This corresponds to the physical fact that the structure of a protein is not affected by the rigid-body translation or rotation of the protein in space.

The most prominent persistent homology-based algorithm for binding affinity prediction, TNet-BP [43], uses a black box algorithm (convolutional neural network) to predict binding affinity directly from these features. While TNet-BP has a high prediction accuracy on a

split of the training set, we find that its accuracy fails to generalize to different datasets, similar to most deep-learning based algorithms (Fig 1). Furthermore, despite the features encoded by persistent homology having geometric relevance, there has been little interpretation of these features and the prediction algorithms have been black box algorithms, even among later works [34,36]. Overall, black box predictions of high stakes targets have been difficult to deploy and reduce to practice [27].

# Results

## Persistence fingerprint: An effective representation of protein-ligand interactions

To overcome the limitations of previous algorithms and develop a machine learning algorithm which is both interpretable and comparably accurate, we describe a one-dimensional representation of the features captured by persistent homology constructed with *opposition distance*, a novel distance function introduced in TNet-BP [43]. The opposition distance between a protein and a ligand atom is their Euclidean distance, while the opposition distance between between atoms of the same affiliation (i.e., both are protein atoms or both are ligand atoms) is infinite. An interpretation of persistent homology under the opposition distance reveals that PATH+ captures the bipartite matching between the protein and ligand at different scales to predict binding affinity (Table 1), which inspires us to term the representation of a protein-ligand complex under opposition distance *internuclear persistence contour (IPC)*

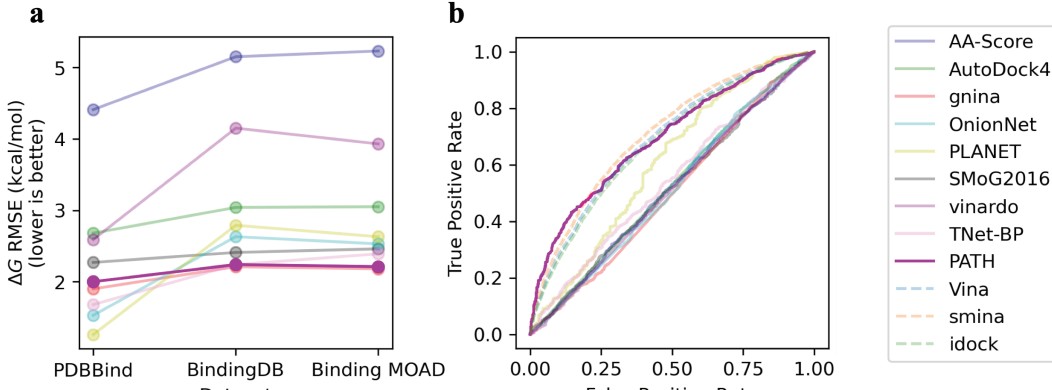

**Fig 1. PATH has state-of-the-art performance versus previous binding affinity prediction algorithms.** [a]PATH+ shows comparable or better performance with less overfitting, as evidenced by a smaller slope, with much less increase in $\Delta G$ RMSEs beyond the training dataset, compared to established binding affinity prediction algorithms spanning a variety of methods. The benchmarked algorithms include physics-based and deep learning algorithms from the famous AutoDock framework (scoring function of AutoDock4 implemented in the AutoDockFR package [68,77], Vinardo [69], GNINA [70]), empirical (AA-Score [71]), knowledge-based (SMoG2016 [72]), and deep learning-based scoring functions (OnionNet [73], PLANET [74]). We believe that PATH+ overfit far less to training dataset than other methods due to the small number of parameters in the sprase regression trees of PATH+. [b]ROC curves of scoring functions benchmarked on the DUD-E dataset show PATH− has state-of-the-art performance in discriminating decoys in the DUD-E dataset. AutoDock4, gnina, and vinardo are all benchmarked as scoring functions. We also plot interpolated ROC curves (dashed) based on AUCs from [75] which benchmarked Vina [78], smina [79], and idock [80] using the full AutoDock framework. The only algorithms with non-diagonal ROCs are PATH− (AUC=0.696), and the three scoring functions tested with the full AutoDock framework: Vina (AUC=0.69), Smina (AUC=0.71), and Idock (AUC=0.68). (Full results in numerical tables in Sect E of S1 Text.)

**Table 1. The 10 features captured by persistence fingerprint.** The source of each feature is represented by a 4-tuple, consisting of [a]the element of protein atoms used in the IPC, [b]the element of ligand atoms used in the IPC, [c]the dimension of the homology group where the IPC is derived from, and [d]the interval (or bin) where the IPC is integrated over to yield the value of this feature. For example, the first row describes that the first component of the vector is calculated by integrating an IPC over the interval [9.5,10.0], where the IPC is constructed using the carbon atoms of the protein and the carbon atoms of the ligand, and the persistence of homology groups of dimension 1 is measured.

| Protein Atom[a] | Ligand Atom[b] | IPC Dimension[c] | IPC Density Bin[d] (Å) |
|---|---|---|---|
| C | C | 1 | [9.5, 10.0] |
| C | C | 1 | [9.0, 9.5] |
| C | C | 1 | [7.0, 7.5] |
| C | C | 1 | [4.0, 4.5] |
| N | C | 1 | [10.0, 10.5] |
| N | C | 1 | [8.0, 8.5] |
| C | N | 0 | [7.5, 8.0] |
| C | N | 0 | [8.5, 9.0] |
| C | O | 0 | [6.5, 7.0] |
| C | S | 0 | [5.0, 5.5] |

(Section Internuclear Persistence Contours (IPCs)). This interpretation also provides structural insights on the features that other persistent homology-based binding affinity prediction algorithms [34,36,43] potentially capture.

From IPCs, we introduce a new representation for protein-ligand interactions, *persistence fingerprints*, which are low dimensional feature subsets that were iteratively refined from a set of persistent homology features inspired by the TNet-BP algorithm [43] and encoded using IPCs. Validation of the generalization power of persistence fingerprints on two large protein-ligand binding databases (Binding MOAD [44–47] and BindingDB [48,49]), which are disjoint from the database whence we curated persistence fingerprint (PDBBind [50,51]), shows that persistence fingerprints can accurately predict binding affinity (Fig 2).

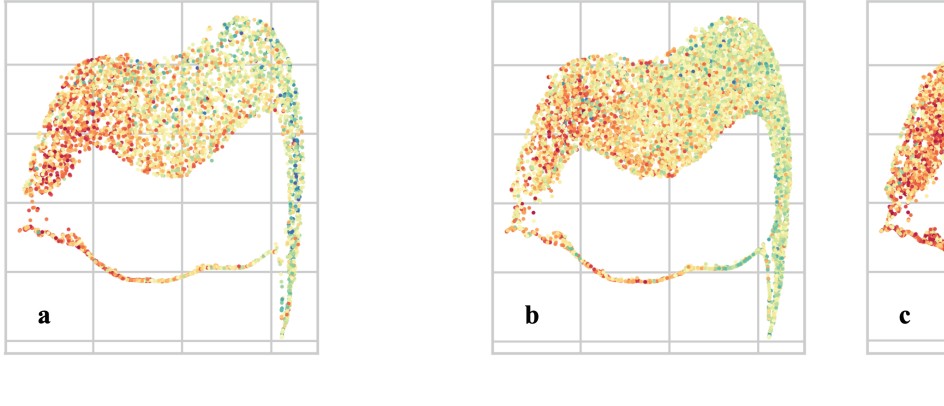

Colormap: Experimental ΔG (kcal/mol)

−3    −4    −5    −6    −7    −8    −9    −10    −11    −12    −13    −14    −15    −16

**Fig 2. Visualization by PaCMAP [52] shows that persistence fingerprint clusters protein-ligand complexes with similar binding affinity reasonably well, even beyond the training dataset (PDBBind v2020 refined set, left panel).** The *x*- and *y*- axes are the dimensionality reduced axes from PaCMAP. The color of each point is the experimental binding affinity of the protein-ligand complex. [a]PaCMAP of the persistence fingerprints of the PDBBind v2020 refined set (training set), [b]Binding MOAD dataset, and [c]BindingDB dataset.

### Persistence fingerprint is highly efficient to compute

We found that persistence fingerprint can be computed highly efficiently, both in CPU time and memory usage.

Previous persistence homology-based binding affinity prediction algorithms generate features that are intractable to run on large protein-ligand complexes: For example, we found that TNet-BP [43] takes over 2 hours to run on protein-ligand complexes with over 8,000 atoms and averages a runtime of 451 seconds per protein-ligand complex on the BioLiP dataset consisting of ∼ 50,000 protein-ligand complexes [53]. Other algorithms that use persistent homology [34,36–38,43,54–56] have similar or worse runtimes, which makes them impractical to use on large protein-ligand complexes.

We propose a leap of an improvement to this by giving a provably accurate approximation algorithm to persistence fingerprint, which allows it be computed in an average of 41 seconds per protein-ligand complex on the BioLiP dataset, a 10-fold improvement in speed over TNet-BP (Fig 3).

Additionally, we offer a provable bound (in terms of computational complexity) to the runtime of computing persistence fingerprint. Let the number of protein atoms be $n$, number of ligand atoms be $m$, and $\omega \approx 2.4$ be the matrix multiplication exponent. Previous binding affinity prediction algorithms that use persistent homology all have computational complexity $\mathcal{O}((m+n)^{4.8})$ or worse [34,36–38,43,54–56]. However, in many biologically relevant protein-ligand complexes [57–59], $n$ can be very large, resulting in unwieldy runtime and space consumption by these algorithms. We found that for any $0 < \varepsilon < 1$, after an $\mathcal{O}(mn\log(mn))$ preprocessing procedure, we can compute an approximation to the persistence fingerprint in $\mathcal{O}(m\log^{6\omega}(m/\varepsilon))$ time, independent of protein size, such that the maximum difference between each component in this approximation and that of the true persistence fingerprint is less than $\varepsilon$. This is an improvement in time complexity by a factor of $\mathcal{O}((m+n)^3)$ over any previous binding affinity prediction that uses persistent homology (Theorem 1). Our ability to create a provably accurate approximation algorithm is primarily due to the small number of features that are employed in the persistence fingerprint.

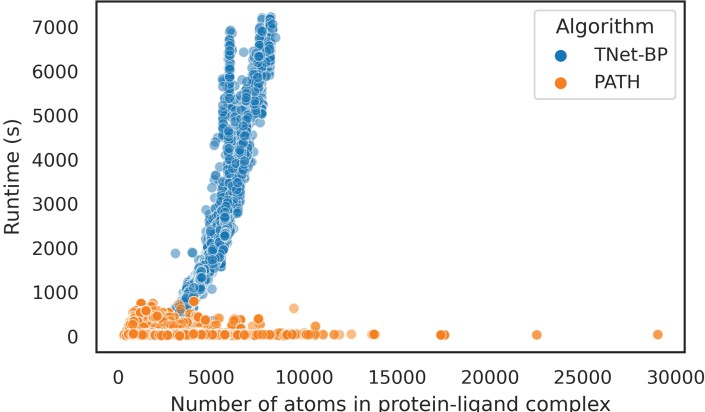

**Fig 3. PATH (with persistence fingerprint) runs significantly faster than TNet-BP, a representative binding affinity prediction algorithm that uses persistent homology, on larger protein-ligand complexes.** The runtime of PATH (shown in orange) is constant with respect of the number of protein atoms in the complex ($n$), while the runtime of TNet-BP is proportional to $n^{7.2}$ asymptotically.

## PATH: A novel algorithm for protein-ligand binding affinity prediction

We propose a novel algorithm for protein-ligand binding affinity prediction, PATH[+] (Fig 4). PATH[+] generalizes beyond the dataset on which it is trained on, making it robust to unseen

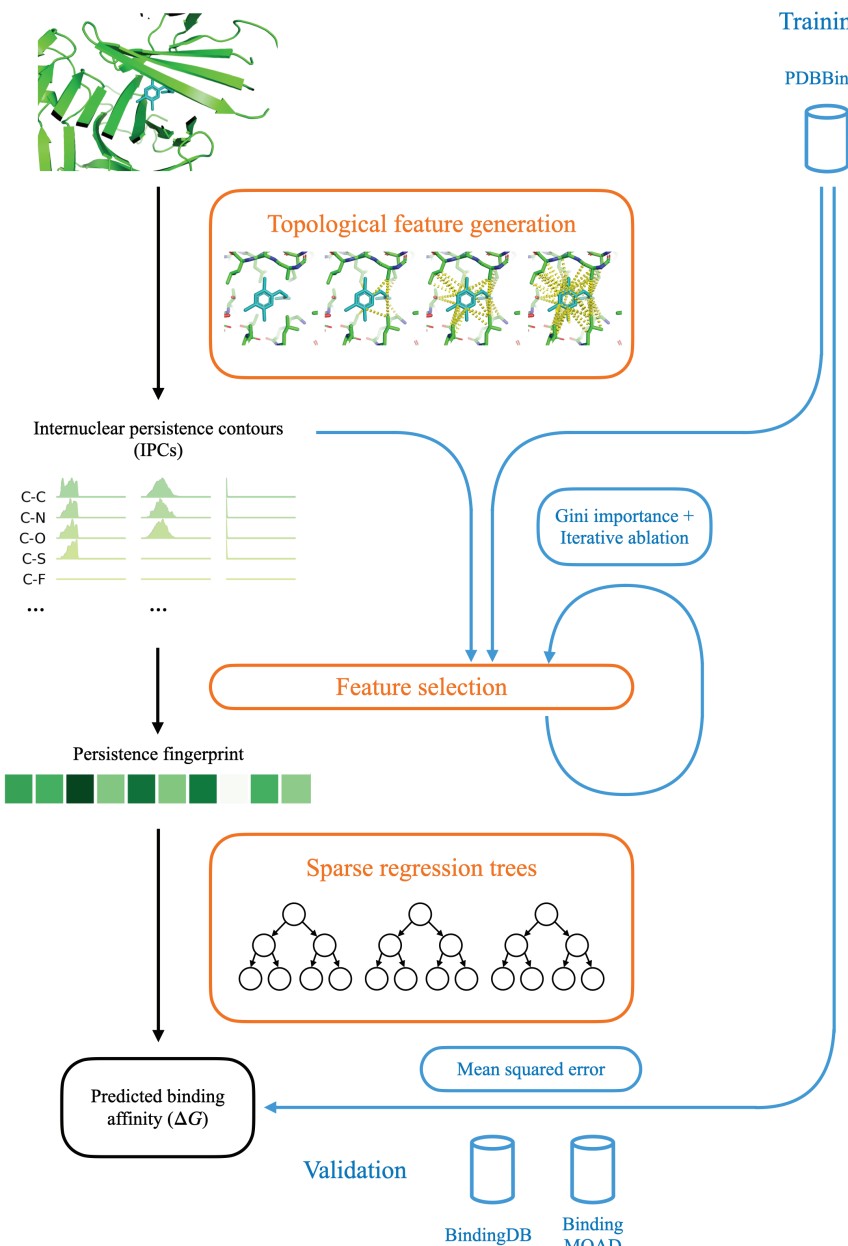

**Fig 4. An overview of PATH[+].** Given a protein-ligand complex, PATH[+] computes internuclear persistence contours (IPCs) using persistent homology, and selects a subset of features into persistence fingerprint, which is then used to predict binding affinity by a sparse set of regression trees (orange). During training (blue), protein-ligand structures with experimentally measured binding affinities from PDBBind are used to derive an optimal set of features for persistence fingerprint and an optimal set of regression trees.

samples and trustworthy to apply to novel targets (Fig 1). We also propose PATH⁻, an algorithm that scores to discriminate binding from non-binding compounds using the persistence fingerprint.

PATH⁺ uses a small ensemble of shallow regression trees to predict binding affinity from persistence fingerprints. PATH⁻ screens for non-binders by using regression trees to score each protein-ligand complex. A *decision tree* is a predictive model represented by a rooted tree, where each internal node represents a test on an attribute, each branch represents the outcome of the test, and each leaf node represents a discrete target class label [60]. In PATH⁺ and PATH⁻, each internal node tests whether a certain element in the persistence fingerprint is larger than a certain threshold. Decision trees are easily interpretable [61], and as a result have been broadly useful in many applications [62,63]. Predictions of individual trees can be easily made by evaluating the tree manually, but the interpretability decreases with the depth of each tree and the number of trees in the ensemble. A *regression tree* is a decision tree whose target values are continuous variables. PATH⁺ and PATH⁻ are both *gradient boosting regressors (GBRs)*, which are ensembles of regression trees iteratively built up based on the error of the previous iteration. Gradient boosting regressors have been used for protein solvent accessibility [64], protein interactions [65], and predicting protein–RNA binding hot spots [66].

A free, open-source implementation of PATH is available in the computational protein software suite OSPREY [67] at https://github.com/donaldlab/OSPREY3.

## PATH is accurate and generalizes across datasets

First, we measured the performance of PATH⁺ versus TNet-BP [43] on the PDBBind v2020 refined set [50]. Due to the lack of a published source code for TNet-BP, we reimplemented TNet-BP's persistent homology feature generation and neural network exactly as described in [43]. We benchmarked both PATH⁺ and our implementation of TNet-BP on a held-out subset of the PDBBind v2020 refined set (519 protein-ligand complexes). Despite following [43] meticulously, we found that our implementation of TNet-BP performs worse than described in the original paper. We report both the performance of TNet-BP from our implementation and the results from the original paper in Table 2. Even when compared to the results of

**Table 2. Performance of TNet-BP and PATH⁺ on the PDBBind v2020 refined set shows that PATH⁺ achieves similar performance with TNet-BP while having three orders fewer features.** For each of the algorithms we have implemented, mean $\pm$ standard deviation is reported for root mean squared error (RMSE) and $R^2$ between predicted $\Delta G$ and experimental $\Delta G$ over 100 random restarts. The RMSE is comparable in magnitude to a hydrogen bond with the atoms N–H $\cdots$ O, which has a molar energy of about 1.9 kcal/mol [76]. We believe that the modest $R^2$ for PATH⁺ is due to the fact that the small number of trees means that PATH⁺ predicts a relatively small number of discrete values, which hurts the $R^2$ metric of PATH⁺ because $R^2$ tracks small differences in predictions.

| Model | *Source code not available* TNet-BP (as reported in [43]) | *Open-source* TNet-BP (our implementation) | *Open-source* PATH⁺ (this paper) |
|---|---|---|---|
| **Number of input features to regressor** | 14472 | 14472 | 10 |
| $\Delta G$ **Root mean squared error (kcal/mol)** | 1.87 | $2.31 \pm 0.11$ | $2.00 \pm 0.05$ |
| $R^2$ | 0.69 | $0.31 \pm 0.04$ | $0.44 \pm 0.04$ |

TNet-BP reported in the original paper [43], PATH$^+$ sacrifices only a slight amount of accuracy in exchange for interpretability, an important characteristic that TNet-BP does not possess. Compared to TNet-BP, PATH$^+$ uses 1,400-fold fewer features, employing a fully interpretable model (Section PATH$^+$ is fully interpretable), achieving an accuracy better than an open-source implementation of TNet-BP, and an accuracy only 7% less (in RMSE) than a closed-source version – only 8% the energy of a hydrogen bond.

Next, we benchmarked PATH$^+$ and PATH$^-$ against 11 established scoring functions for docking on the PDBBind, Binding MOAD, BindingDB (for binding affinity prediction of binders) and a subset of the DUD-E dataset (for decoy prediction). To avoid data leakage, this subset of the DUD-E dataset was chosen to be disjoint from the training dataset of PATH$^-$. The scoring functions we tested include physics- and deep learning-based algorithms from the famous AutoDock framework (scoring function of AutoDock4 implemented in the AutoDockFR package [68], Vinardo [69], GNINA [70]), empirical (AA-Score [71]), knowledge-based (SMoG2016 [72]), and deep learning-based scoring functions (OnionNet [73], PLANET [74]). We note that the AutoDock framework has an advanced pose generation algorithm, which may filter out non-binding conformations before they reach the scoring functions. Therefore, for the decoy prediction task on the DUD-E dataset, we report not only the performance of AutoDock4, gnina, and Vinardo as standalone scoring functions, but also the performance of Vina, smina, and idock benchmarked on DUD-E using the entire AutoDock framework from [75]. Each scoring function was tested with 1 CPU core, 8GB of memory, and 1 hour of compute time. The RMSEs reported on the positive datasets for each algorithm were computed using the subset of protein-ligand complexes where that algorithm returned a prediction. The AUC of negative datasets were computed by considering the protein-ligand complexes where predictions were not returned as either all binders or all nonbinders, whichever yields a better AUC.

PATH$^+$ achieved RMSEs of 2.00, 2.24, 2.21 in $\Delta G$ (kcal/mol) on PDBBind, BindingDB, and Binding MOAD respectively, which is a comparable or better performance with less overfitting compared to the established binding affinity prediction algorithms we benchmarked (Fig 1). The error is comparable in magnitude to a hydrogen bond with the atoms N–H $\cdots$ O, which has a molar energy of about 1.9 kcal/mol [76]. PATH$^-$ has an AUC of 0.696 on predicting decoys from the DUD-E subset disjoint from the training set of PATH$^-$, outperforms the 7 binding affinity prediction algorithms, and performs similarly to the AutoDock algorithms when they are run with the entire AutoDock framework as reported in [75] (Fig 1).

## PATH$^+$ is fully interpretable

Contributions to PATH$^+$'s predictions can be traced to individual atoms in the input structure thanks to the simplicity of persistent homology and the low dimensionality of persistence fingerprint. To our knowledge, PATH$^+$ is the first *interpretable* algorithm that uses persistent homology to predict binding affinity [34,36–38,43,54–56] (Section PATH$^+$).

Finally, to demonstrate the interpretability of PATH$^+$, we inspected the persistence fingerprint and the atoms contributing to persistence fingerprint of two mutant HIV-1 proteases bound to the small molecule inhibitor darunavir. Figs 5 and 6 show the persistence fingerprint and atoms that contribute to a persistence fingerprint component of two mutant HIV-1 protease variants bound to a small molecule inhibitor darunavir (G48V: PDB ID 3cyw [81] & L90M: PDB ID 2f81 [82]). The structural changes induced by the mutation were captured by the persistence fingerprint (Fig 6). Fig 5 highlights a specific region of the protein-ligand complex, showing how a persistence fingerprint component changed from the structural difference. [82] observed a strong hydrogen bond (2.5 Å) to the carboxylate moiety of Asp30

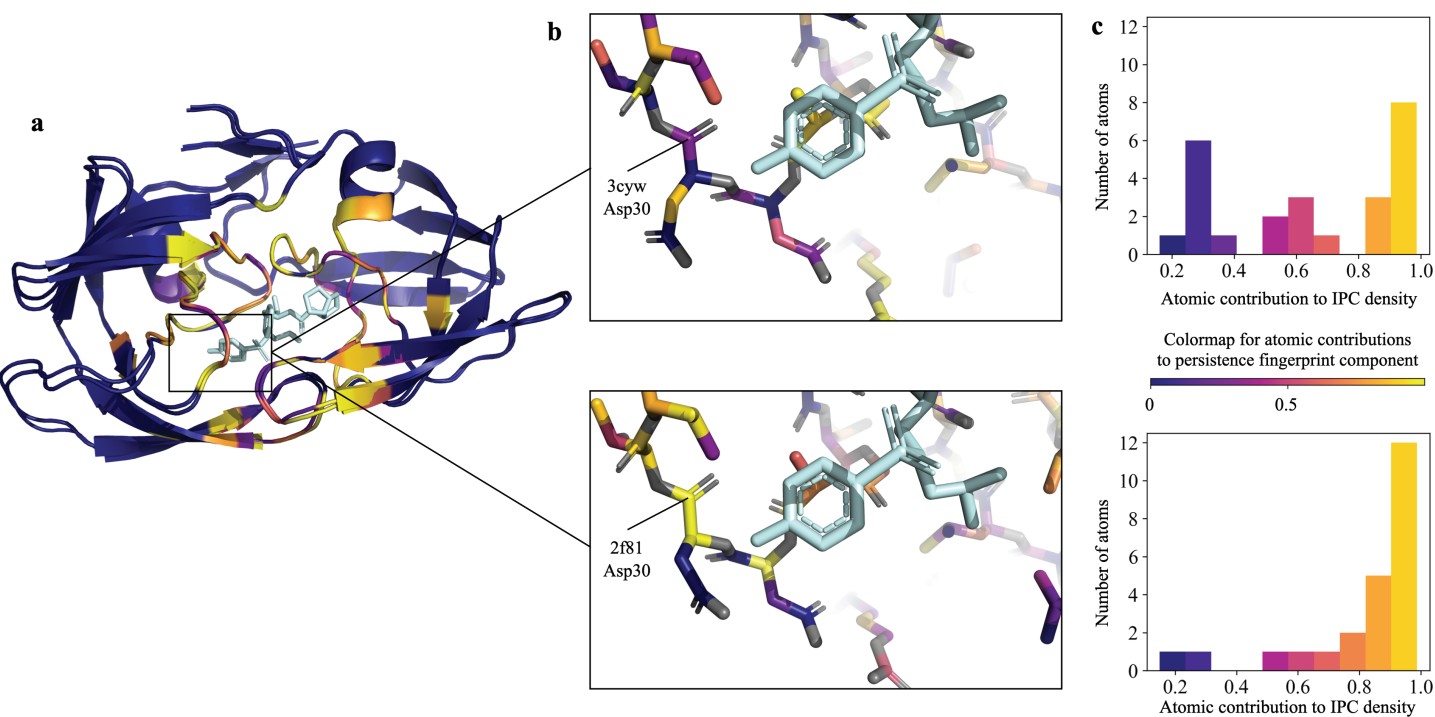

**Fig 5.** **ᵃTwo HIV-1 protease mutants bound to inhibitor darunavir (G48V: PDB ID 3cyw [81] & L90M: PDB ID 2f81 [82]).** Light blue: darunavir. The carbon atoms are colored by their individual contributions (blue through yellow, see legend) to the 2ⁿᵈ component of persistence fingerprint (carbon-carbon IPC density at dimension 1 and bin [9.0, 9.5]). Grey: other protein heavy atoms. **ᵇ**Detail of residues 27-32 for each protease with darunavir. Note change in conformation (and IPC densities) of Asp30. [82] observed a strong hydrogen bond (2.5 Å) to the carboxylate moiety of Asp30. This correlates to Asp30 of the L90M variant contributing highly to the persistence fingerprint component, which obtained a prediction of tighter binding affinity for L90M via the decision trees. **ᶜ**Histograms of atomic contributions of residues 27-32 to the persistence fingerprint shows the carbon atoms of 2f81 in these residues had generally higher contributions to persistence fingerprint.

shown in Fig 5, which correlates with the stronger contributions of backbone carbon atoms to persistence fingerprint in the L90M mutant than in the G48V mutant.

Additionally, to examine the effectiveness of PATH⁺ in predicting binding affinities with different small molecules, we examined PATH⁺ on carbonic anhydrase II bound to two different small molecules, brinzolamide and dorzolamide (Fig 7). Brinzolamide has a flexible methoxypropyl tail, which is noted by [83] to contribute to a stronger binding of brinzolamide (3-fold higher $K_d$) to the enzyme versus dorzolamide. This is predicted correctly by PATH⁺, which shows protein atoms around the methoxypropyl tail contributing highly to the higher predicted affinity of brinzolamide.

## Discussion

Our work highlights interpretability, a previously overlooked aspect in machine learning-based drug design, that persistent homology can bring to embedding biomolecules. The Vietoris-Rips filtration is such a simple construction that a person can effectively compute the persistent homology of a small point cloud by hand. As a result, the features captured by persistent homology can be accurately traced back to the precise atoms that constructed them. Despite the simplicity of their construction, features constructed by persistent homology are sufficient to produce a competent binding affinity prediction model.

The importance of interpretability in algorithms for protein-ligand binding affinity prediction has been recognized by the community, and almost all recent works in binding affinity

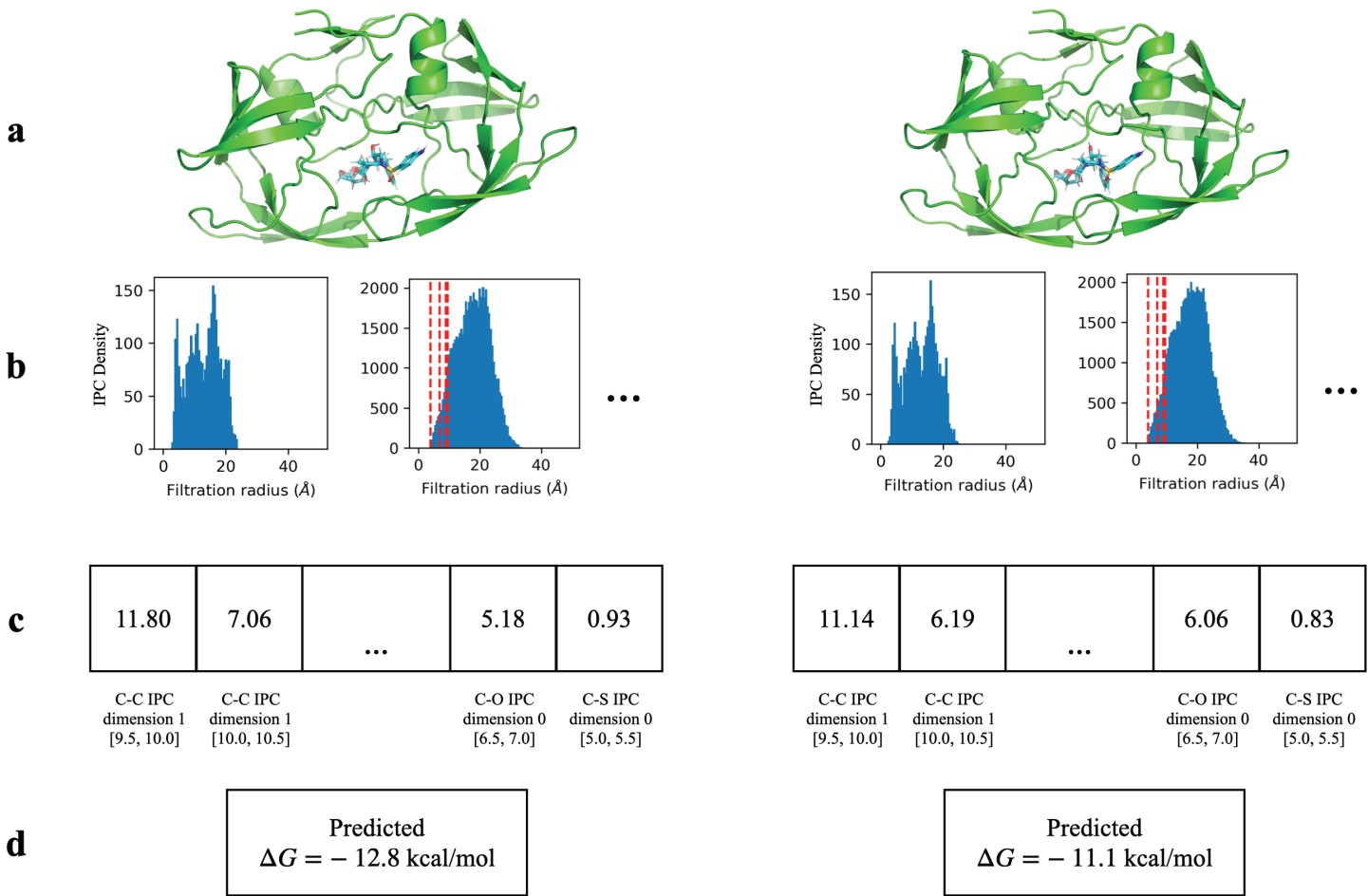

**Fig 6. PATH⁺ correctly predicted a weaker binding affinity for HIV-1 protease with the drug-resistant G48V mutation** (right, experimental $\Delta G$ = −10.6 kcal/mol, PDB ID: 3cyw [81]) bound to darunavir, compared to L90M HIV-1 protease (left, experimental $\Delta G$ = −14.35 kcal/mol, PDB ID: 2f81 [82]) complexed with the same inhibitor. [a] The structure of each complex. [b] The discretized internuclear persistence contour (IPC) of each complex. [c] The persistence fingerprint of each complex. [d] PATH⁺ correctly predicted a weaker binding affinity for the HIV-1 protease with G48V mutation.

prediction [84–88] have made an extra effort to examine the inner workings of their algorithms, via techniques such as neural attention [84,85] and occlusion [86,87]. By terminology of [27], these algorithms are considered *explainable* machine learning algorithms, wherein a second, simpler model is created to explain the primary deep learning model, which is too complicated for humans to understand, *ex post facto*. [27] points out that the simple model is not faithful to what the black box model actually computes and the resulting *post hoc* explanations are often misleading: Saliency maps, which are often used to explain vision models [89], can highlight where the model is looking during a certain decision, but struggle to explain how that decision is made [27]. Effectively explaining black box models, such as language models, is very difficult even for industry AI giants and has been a multi-year research effort in Anthropic and OpenAI [90–93].

PATH is different. PATH is transparent – the persistent homology feature construction and decision trees of PATH can be directly worked out by hand. This allows for direct interpretation of PATH's decision process, which avoids the complications of a *post hoc* explanation. We present PATH not only as an algorithm in its own right, but also to highlight the potential for

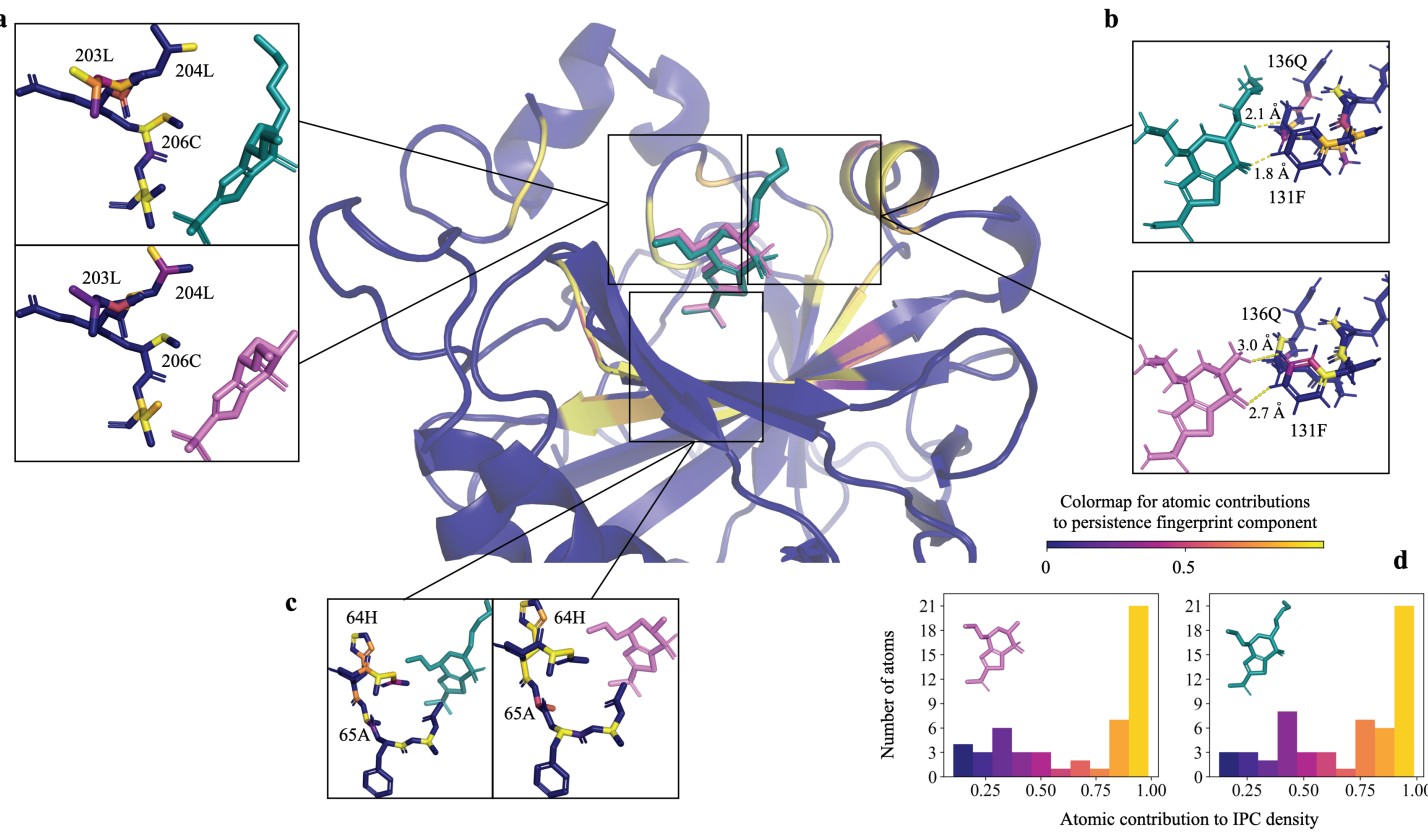

**Fig 7. PATH+ explains tighter binding of carbonic anhydrase II by brinzolamide.** Carbonic anhydrase II bound with two inhibitors: brinzolamide (green sticks, PDB ID 4m2r) dorzolamide (pink sticks, PDB ID 4m2u) and [83]. [83] noted that the flexible methoxypropyl tail of brinzolamide could make favorable interactions with the residues of carbonic anhydrase II, resulting in a 0.61 kcal/mol lower measured $\Delta G$ in brinzolamide complex than in the dorzolamide complex, which corresponds to a 3-fold improvement in $K_d$. This corresponds to a 17% stronger contribution of residues around brinzolamide than dorzolamide (residues 62-67[c], 131-136[b], 203-207[a]) to the persistence fingerprint[d], which contributes to prediction of a tighter binding of brinzolamide than dorzolamide by 0.37 kcal/mol lower $\Delta G$ by PATH+. Small changes in $K_d$ (less than one order of magnitude) have been difficult to correctly predict previously, but nevertheless can have great clinical importance [7]. Furthermore, [b] shows atomic distances (in Å) between the closest protein and ligand hydrogen ($^1$H) atoms. The $^1$H-$^1$H distances between brinzolamide and the carbonic anhydrase II PDB model (4m2r) could be close enough for physics-based methods to predict a clash based on this static structure, even though the clash may not persist when dynamics is considered. Based on this observation, we hypothesize a mechanism through which IPC robustly captures binding activity, elaborated in Section Discussion.

persistent homology to build interpretable algorithms in structural biology. In PATH+, only a tiny subset of features encoded by persistent homology is needed to achieve comparable performance with previous works, and is the core innovation driving PATH to only require a simple ruleset. The persistence of "holes" can capture bipartite matching of protein and ligand atoms at different spatial scales. Invariance to translation and rotation of its input and stability under small perturbations also make persistent homology advantageous for embedding biomolecules.

As opposed to the 14,472 features in TNet-BP, a previous work using persistent homology, our persistence fingerprint representation has only 10 features. PATH+ having comparable performance to TNet-BP highlights that a lot of the complexity in TNet-BP is unnecessary. Even the extended set of features screened from our first pass of feature selection (Table F in S1 Text) only contains 0- and 1-dimensional persistent homology features constructed with opposition distance, showing other complicated feature constructions in TNet-BP (Table A in S1 Text) were unnecessary. Having three orders of magnitude fewer features not only helps

interpretability, but also mitigates overfitting for the downstream algorithm, which is prominent in deep learning models in the biochemical field due to the curse of dimensionality that arises from data that are naturally high-dimensional. We propose that the generalizability of the features captured by persistence fingerprint means that persistence fingerprint can serve as a feature set for future machine learning models for molecular interaction.

Due to the good generalizability of persistence fingerprints and the high performance of PATH⁻ in discerning decoy compounds (Fig 1), we believe that the features in IPCs and persistence fingerprints (Table 1), while not directly corresponding to biophysical observables, can capture denoised information about binding. The most prominent features captured by persistence fingerprint are the number of protein-ligand carbon atoms with distances 9.5–10, 9–9.5, and 7–7.5 Angstroms (Table 1). We hypothesize that because the protein side chains are flexible, a frozen crystal structure may not capture the ensemble of their motion and structural heterogeneity; Instead of using the forces computed from physics-based principles on protein side chains in the crystal structure, persistence fingerprint learns that the distance from the protein backbone to the ligand is a stronger indicator of protein-ligand interaction. In other words, the probabilities of binding have been effectively marginalized over the side chain ensembles, and then attributed to the more invariant backbone atoms. For example, in Panel B of Fig 7, brinzolamide, which binds to carbonic anhydrase II better than dorzolamide (presumably due to brinzolamide's flexible tail [83]), has hydrogens in the static crystal structure with distances 1.8 and 2.1 Å in the hypothesized active region, with intermolecular inter-hydrogen distances close enough be predicted by physics-based methods as unfavorable clashes. However, the flexibility of both brinzolamide and the phenylalanine side chain in this region could lead to interatomic distances that are favorable for binding in ensemble, which should be bettered captured by the more invariant backbone positions. Other possibilities of information captured by persistence fingerprint include allosteric interactions [94], domain reorientation [95], and solvent mediated interactions [96]. Remarkably, through a subsequent literature review, we discovered the features in persistence fingerprint, which were completely automatically derived, are similar to the "interaction fingerprints" manually constructed in previous works on binding affinity prediction [97,98]. Interpretability of PATH⁺ provides verification of the robustness beyond simply benchmarking on datasets and provides insights on the geometric features important to predicting binding affinity with persistent homology. The ability to pinpoint the precise atoms that contribute to a feature in persistence fingerprint also enables us to visualize the structural changes that drive a change in binding affinity, and justify a highly efficient approximation of persistence fingerprint (Theorem 1). This foregrounds the benefits of an interpretable algorithm. PATH can be further accelerated by using approximation algorithms for persistent homology [99–101].

A current weakness of PATH⁺ is that it struggles to rank protein-ligand complexes with small affinity differences, possibly induced by point mutations, due to the fact that the regression tree ensemble can only output a discrete set of values. However, a previous (uninterpretable) persistent homology-based algorithm reports good results [36] on predicting binding affinity change upon mutation with persistent homology, so we believe an interpretable persistent homology algorithm for this task can be developed with the same approach of PATH⁺, obtaining the added benefits of being fast and generalizable. Furthermore, protein redesign algorithms, such as OSPREY, have been experimentally shown to be reliable at predicting binding affinity change upon mutation [59,67,102,103] and can complement PATH in this use case.

## Methods

### Persistent homology

The *persistent homology transform* of a geometric complex such as a protein-ligand structure is a cosheaf of combinatorial persistence diagrams. Specifically, the persistent homology of a point cloud is obtained as the composition of the birth-death functor and the Möbius inversion functor [104]. In operational terms, persistent homology takes a point cloud with a distance function and computes the *persistence* of "holes" of different dimensions at different spatial resolutions. In the case of protein-ligand complexes, the point cloud consists of the centers of all the protein and ligand atoms (e.g., from the Protein Data Bank [105]), and the pairwise distance is usually the Euclidean distance or in our case, the opposition distance (Eq (2)). There are different *filtration functions* from which persistence fingerprints can be constructed. We describe persistent homology using the Vietoris-Rips filtration, which is employed to construct persistence fingerprints. (A formal definition of persistent homology can be found in Sect A in S1 Text.)

A simplex is the generalization of a filled triangle to higher dimensions. A 0-simplex is a point, a 1-simplex is a line segment, a 2-simplex is a triangle, and so on. Given a point cloud $S$ and a pairwise distance matrix, we can define the Vietoris-Rips complex built on this point cloud with radius $r \in [0, \infty)$ by constructing a simplex for any set of points $\sigma$ whose pairwise distance is at most $r$ [106]. Let $\mathbf{VR}_r(S)$ denote the Vietoris-Rips complex built on $S$ with radius $r$, then

$$\mathbf{VR}_r(S) = \{\sigma \subset S : \text{diam } \sigma \leq r\} \tag{1}$$

where diam $\sigma$ denotes the the supremum over the distances between the points in $\sigma$.

Geometrically, the rank of the $n^{\text{th}}$ homology group measures a topological invariant: the number of $n$-dimensional holes in the simplicial complex. For example, the $0^{\text{th}}$ homology group measures the number of connected components, the $1^{\text{st}}$ homology group measures the number of loops, and the $2^{\text{nd}}$ homology group measures the number of voids. In persistent homology, a sequence of simplicial complexes is built up with respect to an increasing *filtration parameter*, which is the radius $r$ in the case of Vietoris-Rips complex described above, and change in rank of the homology group (i.e., the appearance or disappearance of copies of $\mathbb{Z}$ in the direct sum generating the homology group) is measured [43,107]. Geometrically, persistent homology measures the appearance and disappearance of these topological invariants (Fig A in S1 Text).

One way to represent the persistence of these topological invariants is by *persistence diagrams*. Persistence diagrams represent the birth and death of each invariant as a point $(x,y)$, where $x$ is the filtration parameter at which the invariant appears and $y$ is the filtration parameter at which the invariant disappears [108]. Since there can be varying numbers of topological invariants, a vectorization technique is used to convert persistence diagrams to fixed size vectors to employ machine learning techniques, such as support vector machine, decision tree, and neural networks [108]. All of these require a fixed size input. In the construction of persistence fingerprints, we constructed *internuclear persistence contours (IPCs)*, which is a special case of *persistence images* (see Section Internuclear Persistence Contours (IPCs) for a detailed explanation of IPCs. The definition of persistence images can be found in Sect A.2 in S1 Text.) to vectorize the persistence diagrams. Persistence images are provably stable with respect to input noise [108].

## The TNet-BP algorithm

TNet-BP from the TopologyNet family of algorithms [43] is a previous persistent homology-based algorithm for protein-ligand binding affinity prediction (Table 3). TNet-BP [43] introduced the *opposition distance* ($d_{op}$) between two atoms $a_i$ and $a_j$ as follows:

$$d_{op}(a_i, a_j) = \begin{cases} d(a_i, a_j) & A(a_i) \neq A(a_j) \\ \infty & A(a_i) = A(a_j) \end{cases} \tag{2}$$

where $d(\cdot, \cdot)$ is the Euclidean distance between two atoms and $A(\cdot)$ denotes the *affiliation* of an atom, which is either a protein or a ligand. Note that opposition distance does not satisfy triangle inequality and does not have a clear interpretation by itself. Rather, opposition distance works together with the construction of Vietoris-Rips complexes and persistent homology to capture bipartite matching of protein and ligand atoms at different scales. TNet-BP employed 36 persistent homology diagrams with 36 different subsets of atoms using opposition distance and 4 additional persistent homology diagrams using Euclidean distance. The center of each atom (in 3D) is used as its position in the point cloud.

To vectorize each persistence diagram, TNet-BP [43] created a 200 × 72 array where every row represents the births, deaths and persistences of features in one dimension of a persistence diagram. Each row consists of 200 bins, and the value of each bin is the number of features in the persistence diagram that fall into that bin. Finally, to predict binding affinity, TNet-BP used a convolutional neural network on this vector representation. This algorithm was trained and tested on the PDBBind v2020 refined set, a dataset curated from the Protein Data Bank [105] with protein-ligand complexes with their experimental binding affinities [13,50,51,109].

## Internuclear Persistence Contours (IPCs)

Through feature selection (detailed in Sect B of S1 Text), we found that only the 0- and 1-dimensional persistence diagrams constructed with the opposition distance ($d_{op}$) are necessary for accurate binding affinity prediction. We make the following observations about the Vietoris-Rips filtration constructed with the opposition distance:

1. The 0D homology groups all have birth radius 0. This can be seen by the interpretation of 0D homology groups as the number of connected components: as the filtration radius increases, the number of connected components can only decrease. Given a protein atom and a ligand atom, the appearance of a 0D hole happens when the filtration

**Table 3. Comparison of PATH+ with TNet-BP [43].** PATH+ uses a much lower dimensional input vector than TNet-BP, which allowed the use of an interpretable regression model.

|  | TNet-BP [43] | PATH+ (this paper) |
| --- | --- | --- |
| **Dimensionality reduction method** | Persistence diagrams constructed using opposition distance and Euclidean distance on 40 subsets of atoms | Internuclear persistence contours (IPCs) |
| **Input vector to regressor** | Binned persistence diagrams | Persistence fingerprints refined from IPCs |
| **Input vector dimension** | 14,400 | 10 |
| **Regressor Algorithm** | Convolutional neural network | Sparse ensemble of regression trees |
| **Interpretation?** | No | Yes |

radius $r$ equals 0 and disappearance of a 0D hole happens when $r$ equals the distance between these two atoms.

2. The 1D holes all have death radius $\infty$. The birth radius of a 1-dimensional hole corresponds to distances of bipartite matchings of the protein and ligand atoms (Lemmas 1 and 2 in S1 Text).

Note that each feature captured by persistent homology with opposition distance involves the distance between a protein and a ligand atom, hence the similarity with bipartite graphs. Also note that the persistence of each 0D and 1D homology group constructed with opposition distance can be captured using only a single scalar value (death time for 0D, birth time for 1D), rather than a two-dimensional vector. We term this scalar value the *critical value* of persistence.

This allows us to introduce a new representation of the persistence diagrams constructed with opposition distance. An IPC is a function $\gamma : \mathbb{R} \to [0, \infty)$. Given a 0- or 1-dimensional persistent homology constructed with the opposition distance, we can construct its corresponding IPC by summing Gaussians of a given standard deviation centered at each of its critical values (Fig 8). A precise definition is given in Sect A.3 of S1 Text. In our paper, the standard deviation of the Gaussian is chosen to be 0.1 Å.

IPCs can be discretized by taking the integral of IPC over bins of a fixed width. We call the value of the integral over each bin *IPC density* and the resulting collection of IPC densities the *discretized IPC*. The discretized IPC is a nonnegative real-valued function with respect to the bins. Discretized IPCs are closely related to persistence images [108], but discretized IPCs are derived only from persistent homology whose information can be encoded in one dimension (such as when constructed with opposition distance), as opposed to persistence images which can be derived from any persistent homology construction.

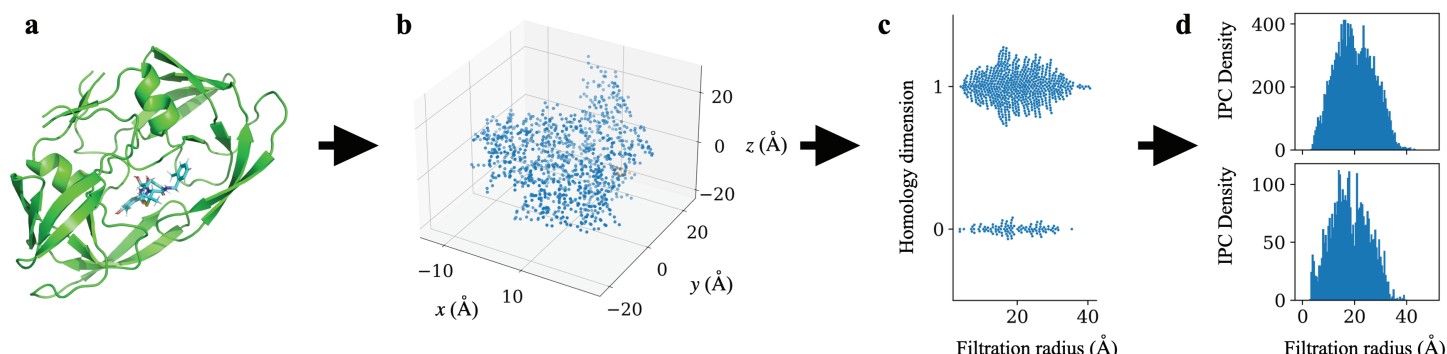

**Fig 8. Construction of internuclear persistence contours (IPCs).** IPCs are constructed for each pair of protein and ligand heavy atoms in the training dataset, and the integrals of IPCs in certain bins are selected into persistence fingerprint. [a]Protein-ligand complex shown as example: the HIV protease (mutant Q7K/L33I/L63I) complexed with KNI-764 (an inhibitor), PDB ID: 1msm [110]. [b]A point cloud is created from subsets of atoms with certain element types in protein and ligand (detailed in Table A of S1 Text). Shown as example: carbon atoms from the protein and carbon atoms from the ligand. [c]Persistent homology is calculated on this point cloud using opposition distance, and the birth filtration radii for 1D homology groups and death filtration radii for the 0D homology groups are collected (see Section Internuclear Persistence Contours (IPCs) for why these suffice). [d]Internuclear persistence contours (IPCs) are constructed by summing Gaussians centered at each of the birth or death radius. The IPCs in PATH are constructed with a standard deviation of 0.1. Two IPCs are shown. Top: carbon-carbon IPC dimension 1. Bottom: carbon-carbon IPC dimension 0.

## Persistence fingerprint

For a given protein-ligand complex, we first constructed 36 pairs of IPCs (0D and 1D) from the 36 subsets of atoms used to construct opposition distance-based persistence diagrams in TNet-BP [43]. Then, we selected the most important components in the IPCs for binding affinity prediction by constructing gradient boosting regressors (GBRs) on the IPCs, identifying the most important features measured by their mean decrease in impurity [111,112] and through an iterative feature ablation procedure [113] (A detailed account of the iterative ablation procedure is given in Sect B of S1 Text). We found that 10 specific components from the discretized IPCs suffice to produce a binding affinity prediction model with comparable performance to TNet-BP. We call the vector made up of these 10 components the *persistence fingerprint*.

**Theorem 1** (Complexity of approximating persistence fingerprint). *Assume there exists a fixed lower bound on interatomic distances in a protein-ligand complex. Let the number of protein atoms be n, the number of ligand atoms be m, and $\omega \approx 2.4$ be the matrix multiplication exponent [114]. For any $0 < \varepsilon < 1$, after an $\mathcal{O}(mn\log(mn))$ preprocessing procedure, we can compute an approximation to the persistence fingerprint in $\mathcal{O}(m\log^{6\omega}(m/\varepsilon))$ time, independent of protein size, such that the maximum difference between each component in this approximation and that of the corresponding element in the true persistence fingerprint is less than $\varepsilon$.*

Proof of Theorem 1 relies on the choice of a weight function for IPCs that decays exponentially, such as the Gaussian. This leads to convergence of any persistence fingerprint component on a ligand atom $l$ for a protein of any size. Then for any $\varepsilon$, there exists a radius $r_\varepsilon$ such that removing all atoms further than $r_\varepsilon$ from $l$ yields an approximation that is at least $\varepsilon$-accurate. A full proof can be found in Sect C of S1 Text. Additionally, an empirical evaluation on a subset of the BioLiP dataset [53] with 45,199 protein-ligand complexes corroborated our asymptotic runtime analysis and achieved an average runtime of 41.4 seconds to calculate the persistence fingerprint of a protein-ligand complex and a maximum approximation error of $\varepsilon = 4.8 \times 10^{-7}$, where $\varepsilon$ is defined as in Theorem 1 (Details can be found in Sect C.4 of S1 Text).

As the value of a persistence fingerprint component can be decomposed into a weighted sum of all the 0D or 1D holes in the protein-ligand complex, the contribution of each protein atom to this persistence fingerprint component can be computed by considering all the holes which contain this given atom. This leads to the interpretation that each component of the persistence fingerprint roughly corresponds to the number of protein-ligand atom pairs of certain elements at a certain distance (Fig 9).

## PATH+

While GBRs have been used as regressors in previous persistent homology based binding affinity prediction algorithms, previous models used 20,000 trees [33,34,36,43] and the large number of trees makes these models impossible to interpret. In comparison, PATH+ has only 13 regression trees, which is three orders of magnitude fewer than previous algorithms, all while maintaining a comparable performance (Table 2). The number of trees, tree depth, and learning rate of the GBRs in PATH+ were selected after we measured the performance of GBRs with respect to these three parameters (Fig E in S1 Text shows ablation results with respect to number of trees) and balanced performance (Table 2) with interpretability (Table 2 and Fig 10 show a comparison between PATH+ and TNet-BP, and Table B in S1 Text shows

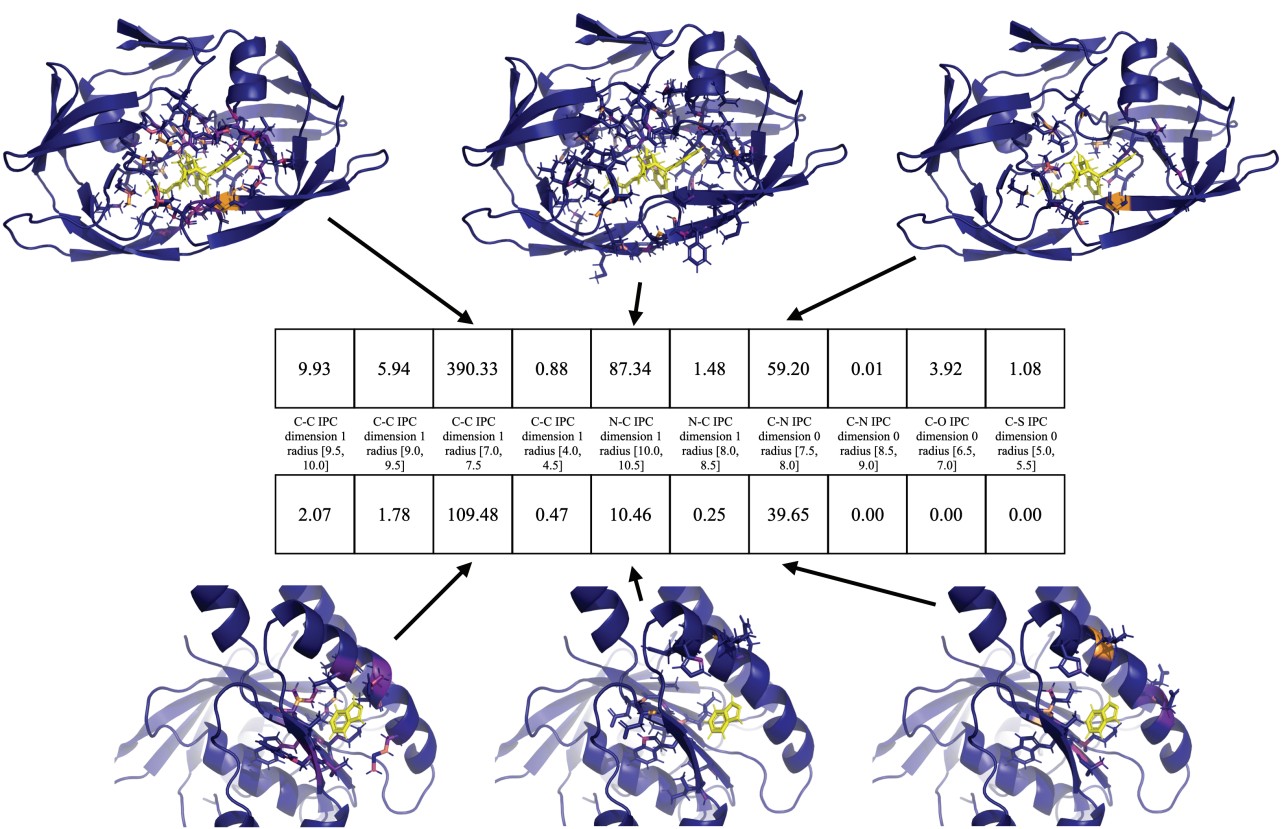

**Fig 9. Two protein-ligand complexes shown with their persistence fingerprint.** <sup>Top</sup>HIV-1 protease in complex with VX-478 (PDB ID: 1hpv [115]). <sup>Bottom</sup>Humanised monomeric RadA in complex with indazole (PDB ID: 4b2i [116]). Contributions to three persistence fingerprint components are shown. The ligand atoms are shown in yellow. Each protein atom is colored according to their contribution to the persistence fingerprint, just like in Fig 5. Each persistence fingerprint component is labeled by the IPC and bin where the IPC is integrated over to yield this component.

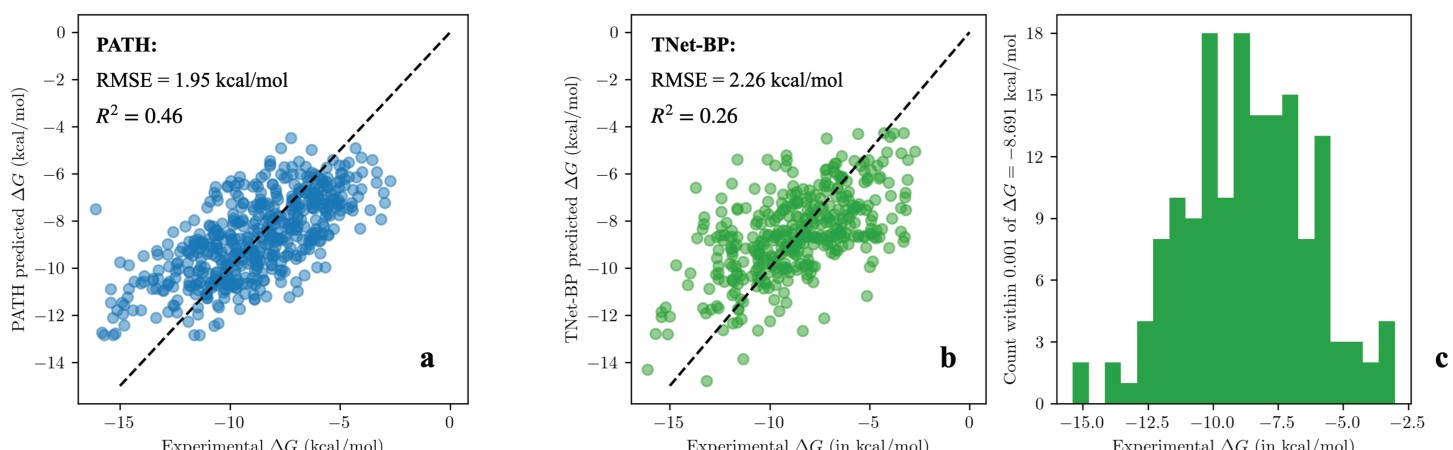

**Fig 10. Scatter plots of PATH⁺ and our implementation of TNet-BP's predictions on a held-out, test subset of PDBBind v2020 refined set for one run (90:10 train:test split ratio, $n_{test}$=519) shows that PATH⁺ produces better predictions, especially on protein-ligand complexes whose binding affinity that deviate significantly from the mean. This highlights PATH⁺'s generalizability.** <sup>a</sup>Predictions of PATH⁺: $R^2$=0.46, $RMSE$=1.95 kcal/mol <sup>b</sup>Predictions of TNet-BP: $R^2$=0.26, $RMSE$=2.26 kcal/mol. <sup>c</sup>To declutter the TNet-BP scatter plot in <sup>b</sup>, we removed 142 data points that are all predicted to have $\Delta G$ within 0.001 kcal/mol of -8.691 kcal/mol by TNet-BP, and instead show the distribution of these points on a separate histogram. The 1-run performances of each algorithm in <sup>a,b</sup> are very close to their average performances over 100 runs in Table 2.

the performance of GBRs with larger inputs and more trees. Additional experiments on alternative regression methods in Sect B.4 in S1 Text confirms that trees-based regressors are optimal on persistence fingerprint). The simplicity of regression trees highlights the representational power of persistence fingerprints. (The precise set of decision trees in PATH$^+$ can be found in Sect D.1 of S1 Text.)

## PATH$^-$

Based on the intuition that binding is a mostly local interaction hence mostly driven by local structures, and the observation that persistence fingerprints from PATH$^+$ (Table 1) all have IPC radius less than 11 Å, we hypothesize that the components of the discretized IPCs that are important to distinguishing between active and decoy compounds are also within a certain radius of the ligand. Therefore, to construct PATH$^-$, we trained a gradient boosting regressor on the 36 pairs of discretized IPCs, the same set that was used in persistence fingerprint of PATH$^+$ (Section Persistence fingerprint), constructed from 551 active and 20227 decoy compounds with 70 proteins from the DUD-E dataset, where only protein atoms within 15 Å from the ligand were used to construct the IPCs. Removal of atoms beyond 15 Å from computing IPCs yields extremely low error in computing the persistence fingerprint for PATH$^+$; hence, we expect the error for important components in discretized IPC due to this approximation to be low as well. Using this approximation, PATH$^-$ achieves the same fast $\mathcal{O}(mn\log(mn)) + \mathcal{O}(m\log^{6\omega}(m/\varepsilon))$ runtime as PATH$^+$.

## Conclusion

Describing the shapes of biomolecules via topological invariants is a promising direction. We filled an important gap in the field of computational structural biology by designing an interpretable vector space with only 10 dimensions for describing protein-ligand interactions, which we call persistence fingerprint. This reduces the dimensionality of the machine learning problem by over three orders of magnitude, opening the door to efficient interpretable algorithms. We showed that the discriminating power of persistence fingerprint generalizes beyond the training dataset. We provided an interpretable algorithm (PATH$^+$) effective at predicting protein-ligand binding using persistence fingerprints. To our knowledge, PATH$^+$ is the first interpretable algorithm for binding affinity prediction using persistent homology, while previous algorithms all resorted to black box models for regression. Despite using three orders of magnitude fewer features, PATH performed with comparable accuracy (only an approximately 7% larger RMSE on the PDBBind v2020 refined set) to TNet-BP, a previous state-of-the-art algorithm that uses persistent homology information for protein-ligand binding affinity prediction. Because the persistence fingerprint we constructed has very few dimensions, we could visually demonstrate that the features measured by our model correspond to biochemically relevant features in the HIV-1 protease-darunavir complex and the carbonic anhydrase II-brinzolamide complex. We also provide the algorithm PATH$^-$, which uses internuclear persistence contours to effectively discriminate between binders and non-binders. We believe PATH will improve existing structure-based drug design pipelines, provide insights in future ones, and enable a novel representation of protein-ligand interactions for future algorithms.

## Supporting information

**S1 Text. This provides additional information to substantiate the claims made in the main paper.** Sect A details the precise definition of persistent homology and the construction of

IPCs, which are the inputs to the regression trees of PATH$^+$ and PATH$^-$. Sect B details the process by which we curated the features used to construct persistence fingerprint from persistence images, and justifies our choice of hyperparameters, such as the number of features and the number of regression trees. Sect C explains the high complexity that is obtained by naively computing IPCs, and the high complexity of previous binding affinity prediction algorithms based on persistent homology (Sect C.1). It also elucidates a fast and provably $\varepsilon$-accurate approximation for persistence fingerprint, which (without our fast approximation algorithm) would naively have inherited the high computational complexity of IPCs. Sect D shows the decision trees of PATH$^+$. Sect E shows the results of benchmarking PATH$^+$ and PATH$^-$ against previous binding affinity prediction algorithms in numerical tabular form, to complement the plots made in main MS Fig 1. Sect F lists the top 77 features selected by the highest mean decrease in impurity in the 120 initial persistence images constructed in Sect B.1.

(PDF)

## Acknowledgments

We thank Graham Holt for his suggestions on initial results, and curation of protein-ligand structures. We thank Henry Childs, Kiran Kanekal, Tomás Lozano-Pérez, Carlo Tomasi, Ron Parr, and Pankaj Agarwal for detailed feedback. We thank Cynthia Rudin, Eric Chen and all members of the Donald research group for proofreading drafts.

## Author contributions

**Conceptualization:** Yuxi Long, Bruce R. Donald.

**Data curation:** Yuxi Long, Bruce R. Donald.

**Formal analysis:** Yuxi Long, Bruce R. Donald.

**Funding acquisition:** Bruce R. Donald.

**Investigation:** Yuxi Long, Bruce R. Donald.

**Methodology:** Yuxi Long, Bruce R. Donald.

**Project administration:** Bruce R. Donald.

**Resources:** Bruce R. Donald.

**Software:** Yuxi Long, Bruce R. Donald.

**Supervision:** Bruce R. Donald.

**Validation:** Yuxi Long, Bruce R. Donald.

**Visualization:** Yuxi Long, Bruce R. Donald.

**Writing – original draft:** Yuxi Long, Bruce R. Donald.

**Writing – review & editing:** Yuxi Long, Bruce R. Donald.

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
