## [Decision Letter · Decision Letter 0]

6 Apr 2025

PCOMPBIOL-D-25-00260

Predicting Affinity Through Homology (PATH): Interpretable Binding Affinity Prediction with Persistent Homology

PLOS Computational Biology

Dear Dr. Donald,

Thank you for submitting your manuscript to PLOS Computational Biology. After careful consideration, we feel that it has merit but does not fully meet PLOS Computational Biology's publication criteria as it currently stands. Therefore, we invite you to submit a revised version of the manuscript that addresses the points raised during the review process.

Please submit your revised manuscript within 30 days Jun 06 2025 11:59PM. If you will need more time than this to complete your revisions, please reply to this message or contact the journal office at ploscompbiol@plos.org. Please include the following items when submitting your revised manuscript:

We look forward to receiving your revised manuscript.

Kind regards,

Jeffrey Skolnick

Academic Editor

PLOS Computational Biology

Nir Ben-Tal

Section Editor

PLOS Computational Biology

**Additional Editor Comments:**

This is a very interesting method for predicting binding affinities. Both the reviewers and we are enthusiastic about this work. Please address the reviewer comments in your revised version.

Congratulations on a very nice contribution.

**Journal Requirements:**

3) Please ensure that all Figure files have corresponding citations and legends within the manuscript. Currently, Figures 8, 9, and 10 in your submission file inventory do not have in-text citations. Please include the in-text citations of the figures.

4) We have noticed that you have uploaded Supporting Information files, but you have not included a list of legends. Please add a full list of legends for your Supporting Information files after the references list. Please cite and label the supplementary tables and figures as “S1 Table” and “S2 Table,” "S1 Figure", S2 Figure" and so forth.

Potential Copyright Issues:

i) Figure 3. Please confirm whether you drew the images / clip-art within the figure panels by hand. If you did not draw the images, please provide (a) a link to the source of the images or icons and their license / terms of use; or (b) written permission from the copyright holder to publish the images or icons under our CC BY 4.0 license. Alternatively, you may replace the images with open source alternatives. See these open source resources you may use to replace images / clip-art:

6) Thank you for stating "The PDBBind dataset can be found at http://pdbbind.org.cn/. The BioLiP dataset (containing BindingDB and Binding MOAD) can be found at https://www.aideepmed.com/BioLiP/weekly.html. The DUD-E dataset can be found at https://dude.docking.org/." Please provide direct links to access each dataset. 

**Reviewers' comments:**

Reviewer's Responses to Questions

Reviewer #1: The authors present the development of a machine learning approach to predict binding affinities for molecular complexes based on their three-dimensional structures. The main algorithm is called PATH+, and its strength is not increased accuracy beyond existing methods, but rather increased generalizability across datasets and a potential for greater interpretability at comparable or better accuracy. PATH+ also has reduced computational complexity compared to other methods. The gains in these dimensions beyond accuracy are noteworthy, and it is important that algorithmic development in the field progress across all of these dimensions, and others as well. PATH+ makes use of advances in computational topology, including persistent homology, and develops useful persistence fingerprints. The authors also present a scoring algorithm PATH–, that distinguishes binders from non-binders. The paper is reasonably well written and reasonably clear. The suggestions below are made for the authors to consider and incorporate at their own discretion.

(1) The standard sections of the paper are exceedingly short. By the end of page 12, one has completed abstract, author summary, introduction, results, and discussion. While some results are presented, much is only alluded to and a great deal is claimed but not shown at the level necessary to make a convincing case to the reader. As but one example, we are told that PATH+ reduces the 14,472 features of a previous method, TNet-BP, which also predicts binding affinity using persistent homology, to only 10 features in PATH+, but there is no real discussion of the differences and what insights we can gather from those differences. Too much of the paper is buried in the overly long methods section.

(2) The claim is made that PATH+ is fully interpretable, but the point is made by argument and not effectively by illustration. We are told this is true because persistence fingerprints are readily interpretable in terms of atomic interactions, and that the decision-tree structure inherent in PATH+ leads to decision-tree-like constructs on atomic interactions, which are therefore interpretable (it reads like a proof). But the real question is whether the interpretations that result from machine learning correspond to those of structure biophysical chemistry, or whether they are something different? And if different, are they at least as useful? Two examples from the text, HIV and CA II, read much more like rationalizations than actual explanations that come from a complete analysis of the ML result. Working out examples in much more detail and much more critically than has been done here, would more meaningfully support the point that the interpretability of PATH+ can lead to trust, which otherwise remains to be seen.

(3) The exact problem statement may be somewhere in the paper, but this reader didn’t find it clearly stated at the start. Are unbound structures the input, bound structures, or both? The answer appears to be bound structures, but is this a requirement? Much has been written about conformational changes on binding and the effects that this can have on binding energy. Does the use of bound structures limit the accuracy that can be achieved?

(4) The authors repeated state that the persistence fingerprint encodes information about topology at different scales, but what about other biophysical features thought to be important for binding? What is encoded in a persistence fingerprint, what additional information could be added by learning over the data, and how does this correspond to what a physics-based modeling approach could capture? Conformational energy changes upon binding? Conformational entropy losses upon binding? Desolvation penalties? The hydrophobic effect? Hydrogen bonds? Quantum-mechanical effects? pH dependence of binding due to titration changes? Temperature dependence? How dependent is the algorithm on knowing accurate titration states? It appears that PATH+ uses a set of concepts that are not directly related to these biophysical concepts but that are none-the-less useful. The extent to which they can be related to biophysical explanations could be interesting to discuss, and might lead to suggestions for additional features for future investigation. The presentation here seems to claim too much and give short shrift to what are truly good questions.

(5) The last sentence of the first paragraph should be reconsidered (“A reliable ranking of docking poses, based on affinity, potency, or other biophysical properties, is essential for accurate SBDD [69].”) The authors don’t seem to believe this based on their introduction of PATH–; they seem to realize that separating the vast majority of non-binders and weak binders from strong binders is valuable, and some reasonable ranking of strong binders is even better. But certainly correct ranking of the weak binders is not at all necessary. I point this out because the topic of the paper is important, and there are many places where a thoughtful argument is necessary to present important subtleties.

(6) In the last sentence of the first paragraph of section 2.1, the word “predictions” should be “prediction”

(7) The opposition distance is not defined until the methods section, although it is used earlier. It would be simple enough to say what it was when first used.

Reviewer #2: The study introduces PATH (Predicting Affinity Through Homology), an interpretable algorithm for predicting protein-ligand binding affinity. It uses a new method to compute persistent homology features for protein-ligand complexes that is more computationally efficient than previous methods and independent of protein size. The algorithm uses internuclear persistent contours (IPCs) and persistence fingerprints to represent protein-ligand interactions. The SI details the persistent homology, construction of IPCs, and feature selection along with the benchmarking of PATH against previous binding affinity prediction algorithms.

Overall, this manuscript is clear, concise, and well-written. The research topic is both interesting and relevant to PLOS Computational Biology. The results indicate that PATH performs comparably to other state-of-the-art methods while exhibiting less overfitting, and it also allows for interpretability. The source code for PATH is available in a GitHub repository. Below are some specific concerns:

The persistence fingerprint used in the PATH algorithm consists of 10 features. While this reduction in dimensionality is presented as an advantage, does increasing the number of features to around 20, where the performance of trained GBRs saturates (see Fig. 14), enhance overall performance? Additionally, how does the computational time increase with the addition of more features?

While the authors demonstrate the algorithm's performance on various datasets, such as PDBBind and BindingDB, it remains unclear how well the method generalizes to different protein families and ligand types.

The authors provided a detailed comparison of PATH with other state-of-the-art approaches, but I did not find the algorithm's relative strengths and weaknesses compared to other deep learning-based affinity prediction methods.

In SI: Fig. 14, the authors presented RMSE versus the number of remaining features, identifying 10 features as the persistence fingerprint with 100 estimators (trees). How these RMSEs vary with 13 estimators as mentioned in Fig.15.

SI: Table 6; R-square values < -1 ?

Please check the order of references, figures, and tables appearing in the main text.

Reference in section heading: 4.2 The TNet-BP algorithm [13]; is not required.

Reviewer #3: The authors descried a topology-based approach, PATH, in structure based drug affinity prediction. They claimed that, in contrast to the more popular deep learning or physics-based approaches, PATH is interpretable and can avoid over fitting and adaptable to new targets, and it runs faster than existing topology based approach too. I must admit that this is not an area that I am familiar, although I do dabble in the drug design area. I do not feel qualified to critique this work.

I do have the following comments.

First of all, is 120+ references necessary? and the numbering doesn’t follow the order they appear in the paper, which makes me think some of these texts may be taken from a thesis.

Deep learning-based approach does suffer from lack of interpretability but there are ways to remedy this to identify residues or chemical groups that are important in binding.

I felt the Results section is a little thin, as they only tested on one dataset and provided one comparison figure (Fig 4), and one case study HIV-1 protease.

**Have the authors made all data and (if applicable) computational code underlying the findings in their manuscript fully available?**

Reviewer #1: Yes

Reviewer #2: Yes

Reviewer #3: Yes

PLOS authors have the option to publish the peer review history of their article (what does this mean?). If published, this will include your full peer review and any attached files.

Reviewer #1: No

Reviewer #2: No

Reviewer #3: No

**Figure resubmission:**
---

## [Editor Report · Decision Letter 1]

10 Jun 2025

Dear Dr. Donald,

We are pleased to inform you that your manuscript 'Predicting Affinity Through Homology (PATH): Interpretable Binding Affinity Prediction with Persistent Homology' has been provisionally accepted for publication in PLOS Computational Biology.

Best regards,

Jeffrey Skolnick

Academic Editor

PLOS Computational Biology

Nir Ben-Tal

Section Editor

PLOS Computational Biology

This revised paper has thoughtfully and appropriately addressed the concerns and comments of the reviewers. This revised version certainly merits publication.

---

## [Editor Report · Acceptance letter]

PCOMPBIOL-D-25-00260R1

Predicting Affinity Through Homology (PATH): Interpretable Binding Affinity Prediction with Persistent Homology

Dear Dr Donald,

I am pleased to inform you that your manuscript has been formally accepted for publication in PLOS Computational Biology. Your manuscript is now with our production department and you will be notified of the publication date in due course.

With kind regards,

Zsofia Freund
